# A Programmable Impedance Tuner with a High Resolution Using a 0.18-um CMOS SOI Process for Improved Linearity

**Younghwan Bae [1], Heesauk Jhon [2,*]**  **and Junghyun Kim [1,*]**

[1] Department of Electronics and System Engineering, Hanyang University, Ansan 15588, Korea; judean@hanyang.ac.kr

[2] Department of Electrical, Information and Communication Engineering, Mokpo National University, Mokpo 530729, Korea

* Correspondence: kindro1@mokpo.ac.kr (H.J.); junhkim@hanyang.ac.kr (J.K.)

**Abstract:** In this paper, a novel coupler/reflection-type programmable electronic impedance tuner combined with switches that were fabricated by a 0.18-um complementary metal–oxide–semiconductor (CMOS) silicon-on-insulator (SOI) process is proposed for replacement of the conventional mechanical tuner in power amplifier (PA) load-pull test. By employing the multi-stacked field-effect transistors (FETs) as a single-branch switch, the proposed tuner has the advantage of precise impedance variation with systematic and magnitude and phase adjustment. Additionally, it led to high standing wave ratio (SWR) coverage and a good impedance resolution with a high power handling capability. Furthermore, the double-branch based on multi-stacked FET was applied to switches for additional enhancement of the intermodulation distortion (IMD) performance through the mitigated drain-source voltage of the single-FET. Drawing upon the measurement results, we demonstrated that SWR changed from 2 to 6 sequentially with a 12–15° phase angle step over a mid/high-band range of a 1.5–2.1 GHz band for 3G/4G handset application. In addition, the PA load-pull measurement results obtained using the proposed tuners verified their practicality and competitive performance with mechanical tuners. Finally, the measured linearity using the double-branch switch demonstrated the good IMD3 performance of −78 dBc, and this result is noteworthy when compared with conventional electronic impedance tuners.

**Keywords:** CMOS SOI process; impedance tuner; intermodulation distortion; multi-stacked FET switch; programmable

## 1. Introduction

Mechanical impedance tuners are often used in the power amplifier (PA) load-pull test which extracts characteristics such as power, gain, and linearity in handset PA applications because of its high performance and reliability. However, despite the excellent performance of the mechanical tuner, its weak points, such as its slow operating speed, high cost, and bulky size, greatly increase the development time and cost in PA applications. Therefore, there is a strong demand for improved impedance tuners from mobile phone makers. Various coupler/reflection-type electronic impedance tuners that are fast, cost-effective, and small-sized with a programmable operation have been conceptually reported as an alternative of the conventional mechanical tuner [1,2]. However, the effectiveness of the reported electronic tuners for a PA load-pull test should be carefully verified with the following performance requirements: (1) operating frequency; (2) programmable operation; (3) standing wave ratio (SWR) coverage; (4) impedance resolution; (5) power handling capability;

and (6) linearity performance. To increase the limited SWR coverage (=2 or 6) and impedance resolution of the PIN diode-type tuner [1], the Varactor diode-type tuner has been reported, with a performance of SWR ≤ 8 [2]. However, the Varactor diode-type tuner exhibits considerable non-linearity behavior in a high power condition due to the increased non-linearity characteristic of Varactor diode devices. Therefore, the Varactor diode-type tuner inevitably has a low power handling capability and poor intermodulation distortion (IMD) performance, despite its high SWR coverage and impedance resolution. An RF complementary metal–oxide–semiconductor (CMOS) silicon-on-insulator (SOI) switch that has a multi-stacked field-effect transistor (FET) configuration can be a good solution, providing a high power handling capability and improved IMD performance for electronic tuner applications because of the mitigated voltage stress of the single-FET.

In this work, a novel programmable electronic tuner using 0.18-um CMOS SOI single-/double-branch multi-stacked FET switches with a high SWR coverage and impedance resolution are proposed, for an improved IMD performance. The proposed tuner, using single-branch 8-stacked FET switches, exhibits a very good IMD3 performance. Furthermore, a dominant non-linearity source of the single-branch type tuner is inspected and the tuner using double-branch 8-stacked FET switches is proposed for an improved IMD performance, without degradation of the other performance. A comparison of conventional tuners and the proposed tuner is summarized in Table 1. The concept of the proposed tuner is presented quantitatively in Section 2, followed by the circuit design in Section 3 and the results in Section 4.

**Table 1.** Programmable impedance tuner and comparison.

| | **Mechanical Tuner [3]** | **Electronic Tuner** | | | |
| --- | --- | --- | --- | --- | --- |
| | | **[1]** | **[2]** | **This work I** | **This work II** |
| Frequency (GHz) | 0.4–4 | 0.82–0.92 | 2.09–2.19 | 1.5–2.1 | 1.5–2.1 |
| Operating type | Programmable | Programmable | Programmable | Programmable | Programmable |
| Switching device | Motor control | PIN diode | Varactor diode + PIN diode | CMOS SOI FET-switch | CMOS SOI FET-switch |
| Topology | Single-stub | Discrete switching | Continuous phasing/attenuating | Discrete switching, coninuous attenuation | |
| SWR coverage | ≤15 | 2 or 6 | ≤8 | ≤6 | ≤6 |
| Impedance resolution | High | Low | High | High | High |
| Power handling capability (dBm) | ≤54 | ≤20 * | ≤13 | ≥26 | ≥26 |
| IMD3 performance (dBc) | N/A | −50 ($P_{in}$ = 30 dBm) | −30 ($P_{in}$ = 13 dBm) | −70 ($P_{in}$ = 30 dBm) | −78 ($P_{in}$ = 30 dBm) |

* Maximum power that can perfom a practical PA load-pull test.

## 2. Concept of the Proposed Tuner

### 2.1. High-Resolution and Variable Magnitude Mechanism

Figure 1a shows a conceptual schematic of the proposed coupler/reflection-type impedance tuner. A conventional PIN diode-type tuner [1] needs many PIN diodes to cause variation of the magnitude part through varying the electrical length (EL) '$\theta_1$ and $\theta_2$' under terminated impedance of the transmission line, where $R_T = 0$ (short), and feasible SWR coverage of the tuner is restricted because of an increased insertion loss of switch elements. Therefore, the terminated impedance variation technique is proposed to achieve magnitude variation and a high impedance resolution without additional switch elements. In Figure 1a, if $(\theta_1 + \theta_2) = 90°$, $Z_0 = 50 \ \Omega$ (characteristic impedance), and $R_T = R$ (where R is any constant), then $|\Gamma_{in}|$ and $\angle\Gamma_{in}$ are

$$\left|\Gamma_{in}\right| = \left|\Gamma(R)\cos(2\theta_1)\right| \tag{1}$$

$$\angle\Gamma_{in} = -2\theta \tag{2}$$

where,

$$\Gamma(R) = \frac{R - Z_0}{R + Z_0} \tag{3}$$

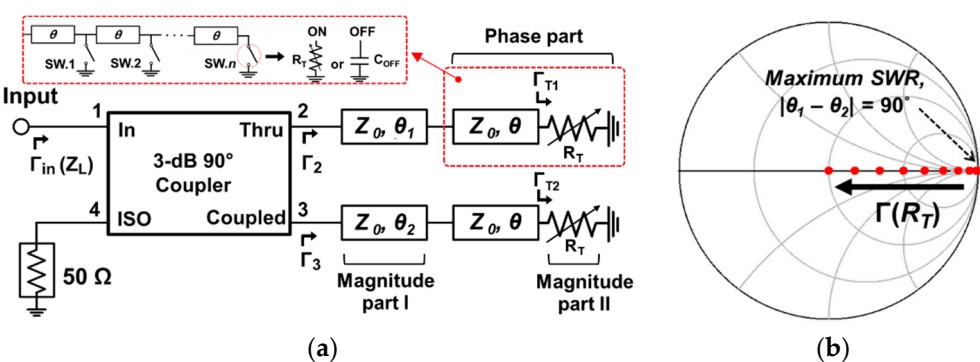

**Figure 1.** (**a**) Conceptual schematic of the proposed coupler/reflection-type electronic impedance tuner. (**b**) $|\Gamma_{in}|$ trajectories obtained by varying $\Gamma_{T1}$ and $\Gamma_{T2}$.

If the magnitude part I is made up of $(\theta_1 - \theta_2) = 90°$ (for maximum SWR) and the variable resistor $R_T$ of the magnitude part II is changed from 0 to 50 $\Omega$ (for decreased variation from maximum SWR) in Figure 1a, magnitude variation of the tuner can be achieved, as shown in Figure 1b. Practically, a phase cell that consists of a transmission line (T/L) and a shunt-type switch ($R_T$ in on-state or $C_{OFF}$ in off-state) are multiply connected in parallel for implementation of the phase part. Therefore, the switches of the phase part cause variation in both the magnitude and phase of the impedance simultaneously.

## 2.2. Power Handling Capability and Linearity of the Tuner

The Varactor diode-type impedance tuner is a good solution for obtaining a high SWR coverage and impedance resolution because of the continuous capacitance variation of a Varactor diode device, while the degraded power handling capability and poor IMD performance in the tuner application are due to the maximized non-linearity of the junction capacitance of the Varactor diode devices (IMD3 = −30 dBc at $P_{in}$ = 13 dBm) [2]. The PIN diode component exhibits good linearity in the forward bias (on-state), while degraded linearity is presented due to the Varacor effect via the junction capacitance of the PIN diode in the reverse bias (off-state). Therefore, the PIN diode-type tuner displays a limited power handling capability and IMD performance (IMD3 = −50 dBc at $P_{in}$ = 30 dBm).

For the case of the CMOS SOI switch with a single-branch structure based on multi-stacked FET, it has the advantage of improving the power handling capability compared to that of the two cases mentioned above, due to the mitigated drain-source voltage ($V_{ds.Mn}$) of the single-FET, as shown in Figure 2 [4–7]. If the shunt branch is in the off-state condition, single-FETs are considered to be a simple capacitive equivalent circuit composed of symmetrical parasitic capacitances ('$C_{gs} \approx C_{gd}$' and '$C_{bs} \approx C_{bd}$') between each node because of the symmetrical drain-source structure and floating gate/body condition, which is implemented by the gate/body resistors ($R_G$, $R_B$, $R_{GC}$, $R_{BC}$) of tens of kilo-ohm or more (see Figure 2). On the other hand, non-linearity in the SOI FET switch-type tuner still occurs, due to adding a half-swing ($V_{ds.Mn}/2$) of the drain-source voltage ($V_{ds.Mn}$) of the single-FET to the gate [8,9] (see Figure 2). The half-swing is induced due to the symmetrical off-state capacitance ($C_{OFF}$) of the single-FET based on a symmetrical drain-source structure and a floating gate/body condition. Eventually, non-linearity of the tuner is generated by the changed gate bias of the single-FET ($V_G + V_{ds.Mn}/2$). Fortunately, the multi-stacked FET switch can restrain the non-linearity of the tuner due to the mitigated drain-source voltage ($V_{ds.Mn} = A_{sig}/n$) and half-swing ($=A_{sig}/2n$) of the single-FET with the increase in the number of stacks, $n$ [10]. Therefore, the multi-stacked FET switch is an excellent solution for obtaining a high-power handling capability and improved IMD performance in tuner applications.

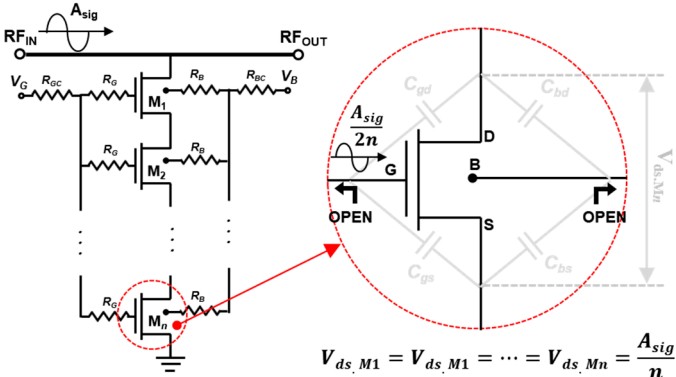

**Figure 2.** Single-branch multi-stacked field-effect transistor (FET) switch configuration and its non-linearity mechanism with the increase in the number of stacks, $n$.

## 3. Circuit Design

Figure 3 shows a detailed schematic of the designed tuner using the CMOS SOI switches. When $R_T = R$, which is a varying on-resistance value of the on-state switches in the loaded-line (LL) part, both terminals of the through and coupled path of the 3-dB coupler are terminated to $\Gamma(R)$ by closing a switch. One LL part consists of 11 T/Ls (EL $\theta_{L1} \approx \theta_{L2} \approx \ldots \approx \theta_{L11} \approx 7°$) and the single-branch switches cover about a 60–70° phase angle, and the upper and lower circuit blocks of the LL part are identical. To achieve phase variation of the impedance, only one switch of the LL part is turned on at a time, but the parasitic parts, such as the LLs and $C_{OFF}$, cause frequency tolerance. Therefore, the last switch (SW.13 in Figure 3) is always turned on and 50 Ω termination is connected to the end of LLs to reduce the parasitic effect. Eventually, phase variation of the impedance can be achieved through the sequential turn on operation of the switches in the LL part (SW.3–13 in Figure 3).

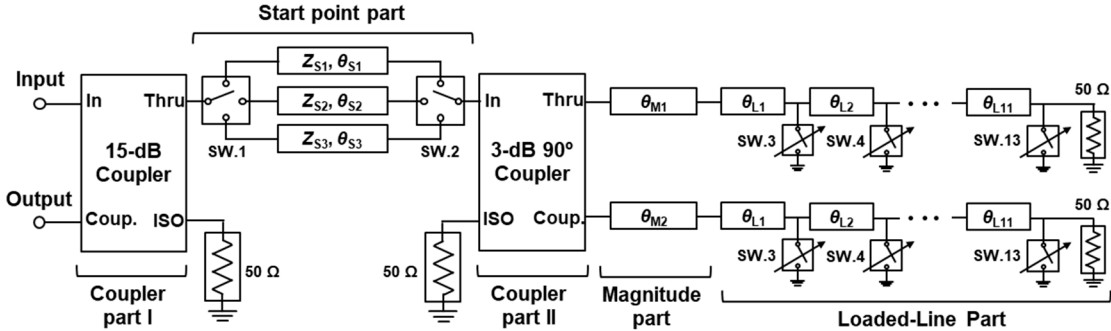

**Figure 3.** Detailed schematic of the proposed tuner using the CMOS silicon-on-insulator (SOI) switches.

The start point (SP) part is employed for the implementation of increased phase variation without added parasitic parts, such as LLs and $C_{OFF}$. The SP part is composed of three fixed T/Ls (EL $\theta_{S1}[Z_{S1}] = 4.5°[45\ \Omega]$, $\theta_{S2}[Z_{S2}] = 68.8°[43.3\ \Omega]$ and $\theta_{S3}[Z_{S3}] = 133°[41\ \Omega]$) and two single-pole triple-throw (SP3T) switches for the selection of a T/L. Therefore, the tuner can cover a 360° phase angle using the SP and LL parts.

The magnitude part consists of the two fixed T/Ls (EL $\theta_{M1} = 23.4°$ and $\theta_{M2} = 118°$), in order to implement the fixed maximum SWR. Next, magnitude variation is caused by the maximum SWR to minimum SWR, by varying the on-resistance of the turned on switch in the LL part (see Figure 1b). Additionally, the 15-dB coupler is connected to the coupler/reflection-type tuner for producing the composition of the two-port network and constant insertion loss of the tuner.

Figure 4a shows the simulated third input intercept point (IIP3)–gate bias ($V_G$) profile of the single-branch 8-stacked FET switch (Figure 2). The gate bias of the on-state switch is controlled to achieve magnitude variation through changing the on-resistance in the LL part. Furthermore, focusing

the non-linearity of the on-state single-branch switch is meaningful because the on-state switch of the LL part under a low gate bias is the dominant non-linearity source of the tuner [11] (the IIP3 of the 8-stacked SP3T switch and off-state single-branch 8-stacked FET switch is always better than that of the on-state single-branch switch of the LL part). Therefore, a degraded IMD3 performance for the tuner is expected under SWR = 2–4. To mitigate the degradation of the expected IMD performance of the tuner, a novel double-branch multi-stacked FET switch is proposed, such as that shown in Figure 4b. To implement tuner SWR = 2–4, the $M_1$ branch-FETs ($M_{1.1}$, $M_{1.2}$, ... , $M_{1.m}$) are turned off in the double-branch switch and the $M_2$ branch-FETs ($M_{2.1}$, $M_{2.2}$, ... , $M_{2.m}$) are short-circuited with the tuned resistor ($R_{TUNE}$), which can mitigate the drain-source voltage of the $M_2$ series single-FETs, as shown Figure 4b. In addition, when the tuner implements SWR = 5–6, the $M_1$ branch-FETs are turned on with off-state $M_2$ branch-FETs. Accordingly, an improved IMD performance for the tuner can be achieved through the mitigated drain-source voltage and/or the half-swing under a low gate bias condition (SWR < 4).

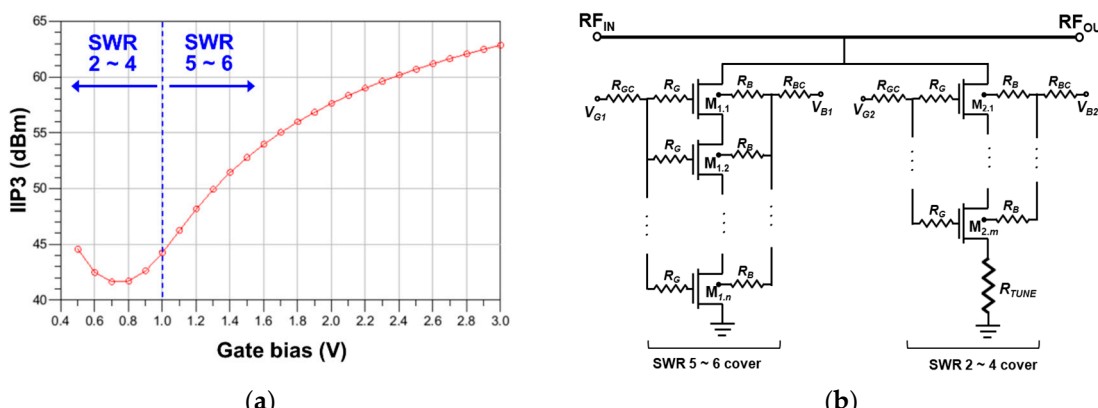

(**a**)　　　　　　　　　　　　　　(**b**)

**Figure 4.** (**a**) Simulated third input intercept point (IIP3) results of the single-branch 8-stacked FET switch (Figure 2) and (**b**) a double-branch multi-stacked FET switch.

## 4. Fabrication and Measurement

Figure 5 shows photos of the fabricated tuners using the single- or double-branch multi-stacked FET switches. The commercial coupled-line chip couplers (3-dB: Anaren's XC1900E-03, 15-dB: AVX's CP0603) and the 10-mil-thick RO4350B PCB (e ~ 3.48, tanD = 0.0037) were employed to achieve low insertion loss and a broad bandwidth (BW). The SP3T switch of the SP part consists of 8-stacked FET series arms (gate width ($W_g$) = 3 mm, gate length ($L_g$) = 220 nm) and shunt arms ($W_g$ = 1 mm, $L_g$ = 280 nm). The SP3T switch has a low on-resistance ($R_{ON}$ ~ 1.6 Ω) and off-capacitance ($C_{OFF}$ ~ 115.5 fF), with a high-power handling capability (>35 dBm). A single-branch 8-stacked FET switch ($W_g$ = 2.5 mm, $L_g$ = 220 nm, $R_{ON}$ ~ 1.92 Ω, $C_{OFF}$ ~ 73.4 fF) was used for the switching operation and variable resistance in the LL part. Additionally, another tuner module using the double-branch 8-stacked FET switch ($W_g$ = 2.5 mm, $L_g$ = 220 nm, $R_{ON}$ ~ 1.92, $C_{OFF}$ ~ 73.4 fF in all branch, and $R_{TUNE}$ = 5 Ω) in the LL part was fabricated to improve the IMD3 performance of the tuner under low SWR.

The complexity of the DC bias system in the fabricated tuners was increased because of including many SOI switch devices. Therefore, an automatic control unit (ACU) was fabricated, as shown Figure 6. The ACU consists of a microprocessor control unit (for fast processing), mechanical relay (for biasing without distortion of the signal), and an insensibility part (for providing voltage fine-tuning). In addition, the ACU is connected to the PC through the serial communication interface. Furthermore, the DC supply is controlled using a general-purpose input/output (GPIO) interface. Eventually, the ACU improves the test reliability and reduces the test time simultaneously. The measured impedance distributions are plotted in Figure 7. The measurement exhibited considerably constant

$|\Gamma_{in}|$, with deviations of less than 0.02 at all SWRs and a very uniform $\angle\Gamma_{in}$ distribution, with deviations of less than 5°.

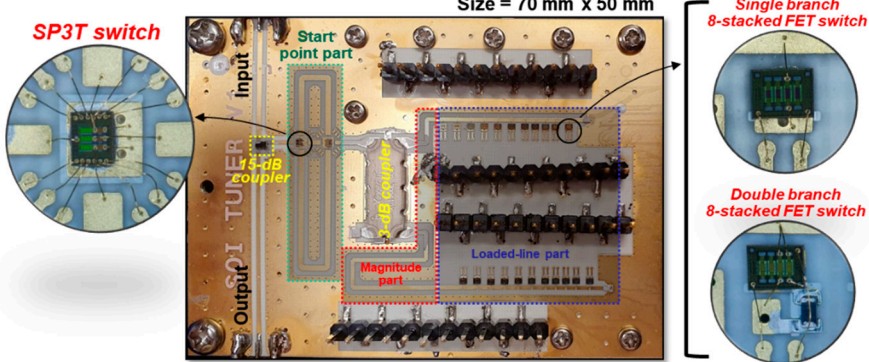

**Figure 5.** Photo of the fabricated impedance tuner using the single-branch 8-stacked FET switch or the double-branch 8-stacked FET switch with $R_{TUNE} = 5\ \Omega$.

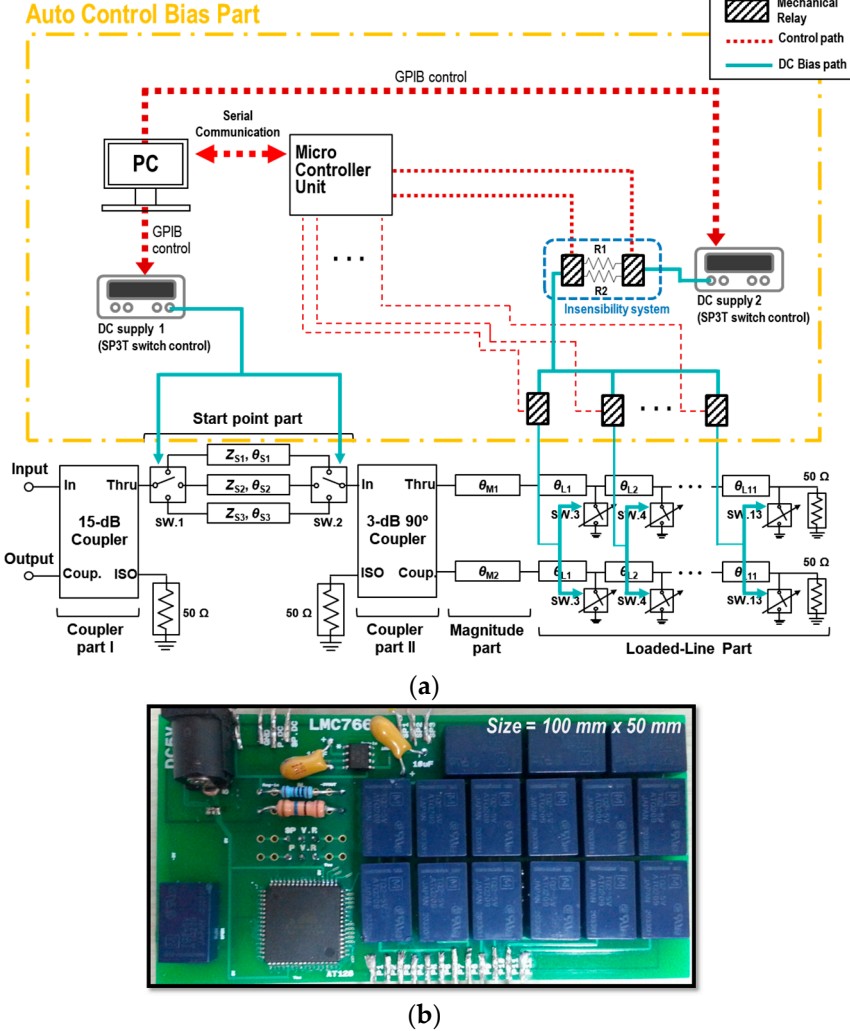

**Figure 6.** DC bias automatic control unit: (**a**) block-diagram and (**b**) photo of the fabricated module.

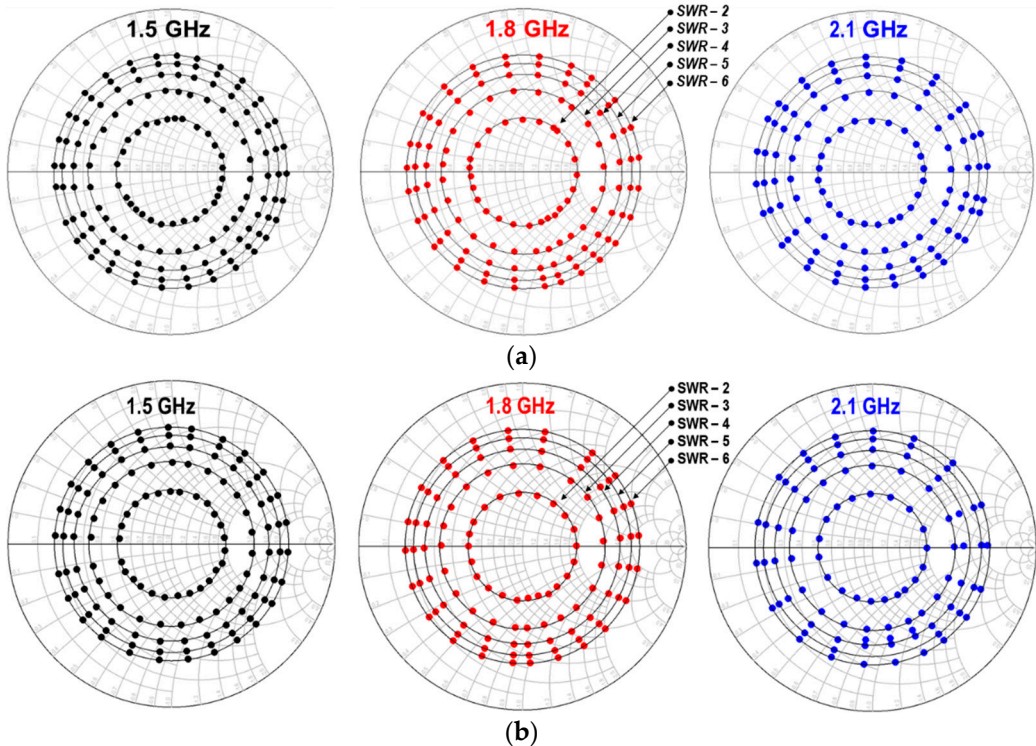

**Figure 7.** Measured impedance distributions of the tuner using (**a**) the single-branch 8-stacked FET switch and (**b**) the double-branch 8-stacked FET switch.

Furthermore, the operation frequency of all the tuners is 1.5–2.1 GHz, and 33.3% BW is achieved, accordingly. In Figure 8a, the measured IMD3 of the tuner using the single-branch 8-staked FET switches was better than −70 dBc at 30 dBm ($f_1$ = 1.8 GHz, $f_2$ = 1.7 GHz, $2f_1 - f_2$ = 1.9 GHz), demonstrating significantly improved results compared to the conventional electronic tuners [1,2]. As expected, the worst IMD3 was occurred under SWR = 4 because of the low gate bias of the on-state single-branch 8-stacked FET switch, as shown Figure 8a. In addition, the IMD3 of the tuner using the double-branch 8-stacked FET switch was improved up to −78 dBc at 30 dBm due to the mitigated drain-source voltage and/or the half-swing of the single-FETs under an SWR lower than 4. To verify the usefulness of the proposed tuners, high-band PA load-pull measurements under a fixed PA output power of 26 dBm were performed, as shown in Figure 9. The proposed tuners were in accord with the mechanical tuner, showing gain deviations of less than 0.5/0.7 dB at all SWR and adjacent channel leakage ratio (ACLR) deviations of less than 1.1/2.4 dB using single-/double-branch-type tuners, over the whole range of phase angles.

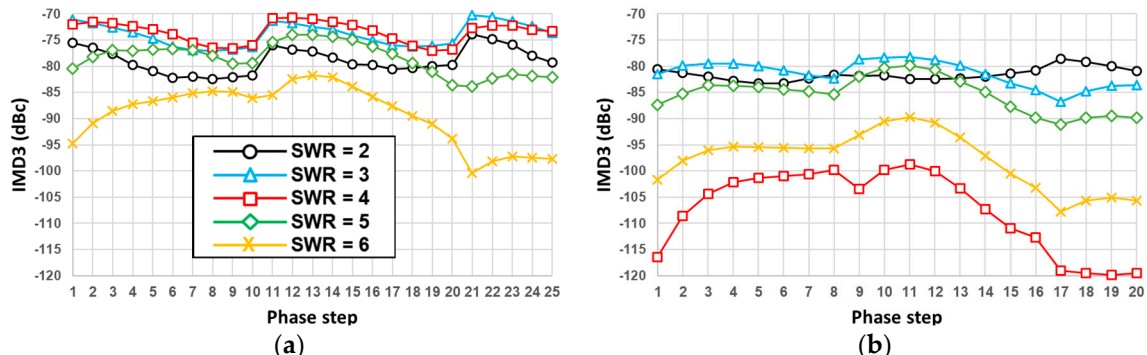

**Figure 8.** Measured third-intermodulation distortion (IMD3) of the tuner using (**a**) the single-branch 8-stacked FET switch and (**b**) the double-branch 8-stacked FET switch.

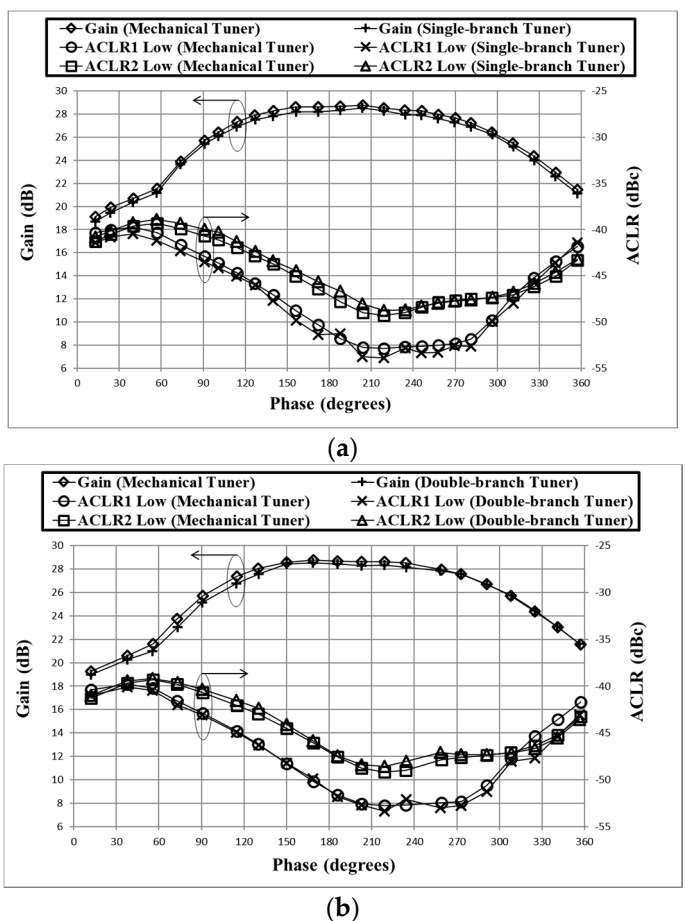

(**a**)

(**b**)

**Figure 9.** Comparison with the mechanical tuner and the proposed tuner with (**a**) the single-branch 8-stacked FET switch, (**b**) the double-branch 8-stacked FET switch, in the power amplifier (PA) load-pull measurement (at $f_0$ = 1.95 GHz and fixed $P_{out}$ = 26 dBm).

## 5. Conclusions

A novel coupler/reflection-type electronic impedance tuner using a 0.18-um CMOS SOI process switch with a two-port network has been proposed for an improved power handling capability and linearity of the tuner. To achieve magnitude variation and a high resolution impedance without a decline of the SWR coverage, the on-state switch of the LL part is used as a variable resistance, and the switch also causes phase variation in the impedance at the same time. In addition, the SP part is employed for increased phase variation of the impedance, without added parasitic parts, such as LLs and $C_{OFF}$. Moreover, the double-branch multi-stacked FET switch is proposed to improve the degraded IMD3 of the tuner under low SWR coverage. Furthermore, the ACU is used with the fabricated tuners to improve the efficiency of measurement. The fabricated tuners use the single- or double-branch switch to implement SWR = 2 to 6 continuously with a 15° phase angle step over mid/high-frequency bands of 1.5–2.1 GHz (BW = 33.3%). Considerably constant $|\Gamma_{in}|$, with deviations of less than 0.02 at all SWRs, and uniform $\angle\Gamma in$, with deviations of less than 5°, are achieved. Furthermore, the PA load-pull measurement is implemented to verify the practical usefulness, the results using the proposed tuners showed good agreement with a mechanical tuner. The measured IMD3 of the tuner using the double-branch switch exhibited a remarkable result that was better than −78 dBc at 30 dBm.

**Author Contributions:** Conceptualization, formal analysis, investigation, and resources, Y.B. and J.K.; methodology, software, validation, and data curation, Y.B.; writing—original draft preparation, writing—review and editing, and visualization, Y.B. and H.J.; supervision, project administration, and funding acquisition, J.K. All authors have read and agreed to the published version of the manuscript.

**Acknowledgments:** This work was supported in part by the Human Resource Program in Energy Technology of the Korea Institute of Energy Technology Evaluation and Planning (KETEP) from the Ministry of Trade, Industry and Energy, Korea (20154030200730).

**Conflicts of Interest:** The authors declare no conflicts of interest.

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
