# Peer review of "A Programmable Impedance Tuner with a High Resolution Using a 0.18-um CMOS SOI Process for Improved Linearity"

_electronics, doi:10.3390/electronics9010007_

Round 1
Reviewer 1 Report
The paper is well written. Simulation results are compared with fabrication measurement results. References are new. I recommend it to be published.
Reviewer 2 Report
This manuscript proposed a programmable electronic impedance tuner using multi-stacked FETs. The manuscript is well organized and the rationale, design and experimental verifications are well executed. However, this manuscript requires extensive English editing to correct grammatical and style issues.
The application for the proposed impedance turner should be explicitly stated right at the beginning of the Abstract.
In Table 1, please specify the tuning technology involved, rather than just "mechanical" or electronics".
Since one of the claims in the proposed multi-stacked FET topology is to achieve a high power handling capability, can you elaborate on this. How does your proposed impedance tuner compare with other state-of-the-art designs?
Here are some (but not all) English issues. There are many more in the manuscript. I recommend seeking professional editing help.
a) In the Abstract, the statement “By employing the multi-stacked field-effect transistor (FET) in single-branch for switch, the proposed tuner has advantages for precise impedance variation with magnitude and phase, systematically.” should be “By employing the multi-stacked field-effect transistors (FETs) as a single-branch switch, the proposed tuner has the advantage of precise impedance variation with systematic and magnitude and phase adjustment.”
b) Also in the Abstract, the sentence “And, it led to give high standing wave ratio (SWR) coverage and good …” should be “It also leads to high standing wave ratio (SWR) coverage and good …”
c) Also in the Abstract, the sentence “In addition, power amplifier (PA) load-pull measurement results using the proposed tuners that verify the practical usefulness showed good agreement with a mechanical tuner.” should be ““In addition, power amplifier (PA) load-pull measurement results using the proposed tuners verified their practicality and competitive performance with mechanical tuners.”
d) First line in the Introduction, the sentence “A mechanical impedance tuner has been usually used for a power amplifier …” should be “Mechanical impedance tuners re often used for in power amplifier …”
Reviewer 3 Report
The paper is well organized and concept, design, and evaluation of the Impedance Tuner technology is satisfactory presented. Some English description can be improved furthermore to make reading more smoothly.
